

# Information theoretic alignment free variant calling

Justin Bedo[1,2], Benjamin Goudey[1,3], Jeremy Wazny[1] and Zeyu Zhou[1,4]

[1] IBM Research—Australia, Carlton, VIC, Australia
[2] Department of Computing and Information Systems, The University of Melbourne, Parkville, VIC, Australia
[3] Centre For Epidemiology and Biostatistics, The University of Melbourne, Parkville, VIC, Australia
[4] School of Mathematics and Statistics, The University of Melbourne, Parkville, VIC, Australia

## ABSTRACT

While traditional methods for calling variants across whole genome sequence data rely on alignment to an appropriate reference sequence, alternative techniques are needed when a suitable reference does not exist. We present a novel alignment and assembly free variant calling method based on information theoretic principles designed to detect variants have strong statistical evidence for their ability to segregate samples in a given dataset. Our method uses the context surrounding a particular nucleotide to define variants. Given a set of reads, we model the probability of observing a given nucleotide conditioned on the surrounding prefix and suffixes of length $k$ as a multinomial distribution. We then estimate which of these contexts are stable intra-sample and varying inter-sample using a statistic based on the Kullback–Leibler divergence.

The utility of the variant calling method was evaluated through analysis of a pair of bacterial datasets and a mouse dataset. We found that our variants are highly informative for supervised learning tasks with performance similar to standard reference based calls and another reference free method (DiscoSNP++). Comparisons against reference based calls showed our method was able to capture very similar population structure on the bacterial dataset. The algorithm's focus on discriminatory variants makes it suitable for many common analysis tasks for organisms that are too diverse to be mapped back to a single reference sequence.

## INTRODUCTION

Many sequencing studies begin by the transformation of raw sequence data to relatively few features, usually single-nucleotide variants. Typically, this is done by aligning the individual sequence reads to a reference genome to identify single nucleotide differences from the reference.

Although straightforward, the genome alignment approach has several shortcomings:

- A suitable reference may not exist; this is especially important for unstable genomes such the anuploid genomes frequently encountered in cancer (*Beroukhim et al.*, *2010*), and also for some organisms with large genetic diversity such as bacteria (*Ochman, Lawrence & Groisman*, *2000*);

Corresponding author
Justin Bedo, cu@cua0.org

- Selecting a reference may be difficult when there is uncertainty about what has been sampled; and
- It performs poorly when a sample contains significant novel material, i.e., sequences that are not simple variations of the reference.

Existing reference-free approaches are either based on assembly (*Li*, *2012*), which possibly introduces misassembly biases, or on searching for structural motifs within a universal de Bruijn graph of all samples (*Peterlongo et al.*, *2010*; *Iqbal et al.*, *2012*; *Uricaru et al.*, *2015*) that correspond to simple variants.

We present a variant calling algorithm to generate features from unaligned raw reads. Rather than attempting to identify all genetic variation within a given set of samples, we instead focus on selected variants that have have strong statistical evidence for their ability to segregate samples in a given dataset. Such variants form useful features for many tasks including genomic prediction of a given phenotype, modelling population structure or clustering samples into related groups.

Our method uses the *context* surrounding a particular nucleotide to define variants. Given a set of reads, we model the probability of observing a given nucleotide conditioned on the surrounding prefix and suffix nucleotide sequences of length $k$ as a multinomial distribution. We then estimate which of these contexts form potential variants, i.e., those that are stable intra-sample and varying inter-sample, using a statistic based on the Kullback–Leibler divergence. Given this list of candidate variants, we call those variants by maximum likelihood of our multinomial model.

Furthermore, we show that the size of the context $k$ can be chosen using the minimum message length principle (*Wallace & Boulton*, *1968*) and that our context selection statistic is $\gamma$-distributed. Consequently, $k$ can be determined from the data and the contexts surrounding variants can be selected with statistical guarantees on type-1 errors.

The utility of variant calling method was evaluated through simulation experiments and empirical analysis of a pair of bacterial datasets and a mouse dataset. Through simulations we showed the method has good power and false positive rate for detecting variants, though the ability to detect rare variants required high depth and large number of samples.

Our empirical results indicated our variants are highly informative for antimicrobial resistance phenotypes on the bacterial datasets and were able to accurately capture population structure. On the mouse dataset, the variants were also found to be good for modelling coat colour. Further investigations of the variants found for the bacterial dataset using a known reference sequence revealed variants associated with boxB repeat regions, a repeat previously used for population structure mapping (*Rakov, Ubukata & Robinson*, *2011*), suggesting the model can generate features for more complex genetic elements. These results suggest the variants are capturing genotypic variation well and can model heritable traits in different organisms. Our proposed method will be of strongest utility when modelling of population structure, phylogenetic relationships or phenotypes from genotype for large scale datasets of organisms with either variable genomes (as is the case for many bacteria), or those lacking a reference genome.

## METHODS

Our variant calling method comprises two steps: modelling the probability that a base is observed in a sample given the surrounding context; and determining which contexts surround variable bases in a population represented by several samples. The former provides a mechanism to call variants in a sample given a set of contexts, and the latter determines the set of contexts associated with variants.

### Variant calling

We consider the case of variant calling directly from a collection of reads. Let random variable $x_{ij}$ taking values in $\{A, C, G, T\}$ denote the $j$th nucleotide of the $i$th read, with $1 \leq i \leq n$ and $1 \leq j \leq m_i$ the number of reads and nucleotides in the read $i$.

**Definition 1** ($k$-context): *The $k$-context around a nucleotide $j$ consists of a $k$-prefix sequence*

$$\pi_k(x_i, j) := [x_{i(j-k)}, x_{i(j-k+1)}, \ldots, x_{i(j-1)}]$$

*and a $k$-suffix sequence*

$$\sigma_k(x_i, j) := [x_{i(j+1)}, x_{i(j+2)}, \ldots, x_{i(j+k)}].$$

*Contexts that consist of only the prefix/suffix sequences are suffix/prefix-free.*

**Definition 2** ($k$-context probability): *The $k$-context probability is the probability of observing a base at a particular position given the context, that is*

$$P(x_{ij} | \pi_k(x_i, j), \sigma_k(x_i, j)).$$

The $k$-context probabilities can be estimated from the data by maximising a pseudolikelihood. Let $f(b, \pi_k, \sigma_k) := 1 + \sum_{ij} [\![ x_{ij} = b \wedge \pi_k = \pi_k(x_i, j) \wedge \sigma_k = \sigma_k(x_i, j) ]\!]$ denote the counts of how often $b$ was observed with $k$-prefix $\pi_k$ and $k$-suffix $\sigma_k$ in the read set $x$, where $[\![ \cdot ]\!]$ is the Iverson bracket. Here the pseudocount encodes a weak uniform prior. The probability density estimate of observing a base $b$ in context $(\pi_k, \sigma_k)$ is then given by

$$\widehat{P}(b | \pi_k, \sigma_k) := \frac{f(b, \pi_k, \sigma_k)}{\sum_{b'} f(b', \pi_k, \sigma_k)}.$$

The suffix/prefix free densities are thus

$$\widehat{P}(b | \pi_k) = \sum_{\sigma_k} \widehat{P}(b | \pi_k, \sigma_k) \quad \text{and} \quad \widehat{P}(b | \sigma_k) = \sum_{\pi_k} \widehat{P}(b | \pi_k, \sigma_k).$$

Given a context $(\pi_k, \sigma_k)$, the base can be called as $\operatorname{argmax}_b \widehat{P}(b | \pi_k, \sigma_k)$, and similarly for prefix/suffix free densities.

## Variant finding

Determining the list of variants consists of determining which contexts $(\pi_k, \sigma_k)$ surround a variable base in our population, then call the base for each variant-defining context and each sample. We consider inter-sample variants and not intra-sample variants; we are interested in finding contexts which define variants that differ amongst samples and are not attributable to noise. In this section, we develop a statistic based on the Kullback–Leibler (KL) divergence that achieves these two points.

Let $\mathcal{X}$ be a set of samples, each consisting of a collection of reads as defined above. For each $x \in \mathcal{X}$, we refer to the $j$th nucleotide of the $i$th read as $x_{ij}$, the number of reads in the sample as $n_x$, and the number of nucleotides in read $x_i$ as $m_{x_i}$. Similarly to the previous section, we denote $f_x(b, \pi_k, \sigma_k)$ as the frequency of observing base $b$ given context $(\pi_k, \sigma_k)$ for sample $x$. As before, a pseudocount is used when estimating $f_x$ to encode a uniform prior.

The KL divergence measure provides a way of quantifying the differences between two probability distributions. We will develop a statistic based upon the KL-divergence that compares the individual sample distributions of nucleotide occurrence for a given context with a global expected distribution. Contexts that significantly diverge from the global expected distribution surround a site which is variant in the population sample.

**Definition 3 (Kullback–Leibler divergence):** *Let $P$ and $Q$ be two discrete probability densities over the domain $\mathcal{Y}$. The Kullback–Leibler (KL) divergence is*

$$P(\cdot) \|_{kl} Q(\cdot) := \sum_{y \in \mathcal{Y}} P(y) \log \frac{P(y)}{Q(y)}.$$

**Definition 4 (Total divergence):** *The total divergence for a given context $(\pi_k, \sigma_k)$ is estimated as the total KL divergence between the samples in the dataset $\mathcal{X}$ and the expected probability distribution given the context:*

$$D_{\mathcal{X}}(\pi_k, \sigma_k) := \sum_{x \in \mathcal{X}} \widehat{P}_x(\cdot | \pi_k, \sigma_k) \|_{kl} Q(\cdot | \pi_k, \sigma_k),$$

*where*

$$\widehat{P}_x(\cdot | \pi_k, \sigma_k) := \frac{f_x(b, \pi_k, \sigma_k)}{\sum_{b'} f_x(b', \pi_k, \sigma_k)}$$

*denotes the probability density estimated for sample $x$ and context $(\pi_k, \sigma_k)$ and*

$$Q(b | \pi_k, \sigma_k) := \frac{\sum_{x \in \mathcal{X}} f_x(b, \pi_k, \sigma_k)}{\sum_{x \in \mathcal{X}, b'} f_x(b', \pi_k, \sigma_k)}.$$

The total divergence statistic is proportional to the expected KL-divergence between a sample and the global expected probability distribution. To see why this statistic is robust to noise, consider the case where variation is due purely to noise. As the noise distribution is independent of sample, it will be well modelled by the expected distribution $Q$ and

therefore the divergence between each sample and $Q$ will be small. Conversely, if variation is due to samples being drawn from two or more latent probability densities, then $Q$ will be an average of these latent densities and divergence will be high.

The next theorem is crucial for determining when a particular divergence estimate indicates a significant divergence from the expected distribution $Q$. Using this theorem, we can use hypothesis testing to select which contexts are not well explained by $Q$. These contexts not well explained by $Q$ are *variant* and we call them as in 'Variant calling.'

**Theorem 5.** *Under random sampling from Q, D follows a $\gamma$ distribution.*

The proof of this theorem is trivial given a well known result regarding the *G*-test (see *Sokal & Rohlf* (*1994*)).

**Lemma 6** *Let $f_x$ be a frequency function and $g := E[f_x]$. The G-test is*

$$G := \sum_{x \in \mathscr{X}} \sum_{b \in \{A,T,C,G\}} f_x(b, \pi_k, \sigma_k) \log\left(\frac{f_x(b, \pi_k, \sigma_k)}{g(b, \pi_k, \sigma_k)}\right).$$

*Under the null hypothesis that $f_x$ results from random sampling from a distribution with expected frequencies $g$, G follows a $\chi^2$ distribution with $3|\mathscr{X}|$ degrees of freedom asymptotically.*

From this lemma, the proof of Theorem 5 follows easily:

**Proof.** $D$ is proportional to the G-test. As the G-test is $\chi^2$-distributed, $D$ is $\gamma$-distributed. □

Clearly our statistic $D$ is very similar to $G$, but has an important property: $D$ is invariant to coverage. As $D$ operates on estimates of the probability rather than the raw counts, changes in coverage are effectively normalised out. This is advantageous for variant discovery as it avoids coverage bias and allows variants to be called for (proportionally) low-coverage areas, if statistical support for their variability in the population exists.

To select contexts, a $\gamma$ distribution is fitted to the data. For the results in our experiments, we used a Bayesian mixture model with a $\beta$ prior over the mixing weights whereby each context could originate from the null ($\gamma$) distribution or from a uniform distribution. The mixing weights were then used to determine if a context is not well supported by the null distribution. Such a model comparison procedure has several advantages and directly estimates the probabilities of support by the data for each context (*Kamary et al.*, *2014*), providing an easily interpretable quantity.

## Choosing context size

The problem of choosing context size $k$ is difficult; if too large then common structures will not be discovered, and if too small then base calling will be unreliable. We propose to choose $k$ using the *minimum message length* principle (*Wallace & Boulton*, *1968*).

Consider a given sample $x$. The message length of a two-part code is the length of the compressed message plus the length of the compressor/decompresser. In our case, the length of the compressed message is given by the entropy of our above probability

distribution:

$$L(x;\widehat{P}(\cdot|\pi_k,\sigma_k)) := -\sum_{ij}\log\widehat{P}(x_{ij}|\pi_k,\sigma_k).$$

The compressor/decompresser is equivalent to transmitting the counts for the probability distribution. This can be thought of as transmitting a $k$ length tuple of counts. Let $N = \sum_i(m_i - 2k)$ be the total number of contexts in the read set (i.e., the total number of prefix and suffix pairs in the data). Thus, $\binom{N+4^{2k+1}-1}{4^{2k+1}-1}$ count distributions are possible amongst the number of total prefix and suffix pairs ($4^k \times 4^k = 4^{2k}$ distinct prefix/suffix pairs, and 4 possible observable bases), giving a total message length of

$$ML(x;\widehat{P}(\cdot|\pi_k,\sigma_k)) := L(x;\widehat{P}(\cdot|\pi_k,\sigma_k)) + \log\binom{N+4^{2k+1}-1}{4^{2k+1}-1}.$$

Approximating the R.H.S using Stirling's approximation and dropping constant terms yields

$$ML \overset{\sim}{\propto} 2L(x;\widehat{P}(\cdot|\pi_k,\sigma_k)) + 2\log(4)\left(k - (1+2k)4^{1+2k}\right) \\ + \left(2\left(4^{1+2k}+N\right)-1\right)\log\left(N+4^{1+2k}\right).$$

For suffix free densities the message length simplifies to

$$ML(x;\widehat{P}(\cdot|\pi_k)) := L(x;\widehat{P}(\cdot|\pi_k)) + \log\binom{N+4^{k+1}-1}{4^{k+1}-1} \\ \overset{\sim}{\propto} 2L(x;\widehat{P}(\cdot|\pi_k)) + \log(4)\left(k - 2(1+k)4^{1+k}\right) \\ + \left(2\left(4^{1+k}+N\right)-1\right)\log\left(N+4^{1+k}\right),$$

and similarly for prefix free.

## Prefix/suffix free contexts

The method we have presented so far has been developed for any contexts defined by any combination of prefix and suffix. The question of whether prefix/suffix-free contexts or full contexts (both prefix and suffix) naturally arises. The decision depends on the type of variants of interest: using full contexts will restrict the variants to single nucleotide variants (SNV), while one sided contexts allow for more general types of variants such as insertions and deletions. Full contexts also have less power to detect variation caused by close-by SNVs; two SNVs in close proximity will create several different contexts when modelling with both prefixes and suffixes. It is also worth remarking that the choice between prefix and suffix free contexts is immaterial under the assumption of independent noise and sufficient coverage. Thus, our experiments concentrate on suffix-free contexts as it is the more general case.

## Reference-based variant calling

To compare the ability of our proposed method to a reference-based approach, we have processed all datasets using a standard mapping-based SNP calling pipeline. Using

SAMtools v1.2-34, raw reads from each sample were mapped to the relevant reference sequence and sorted. The mapped reads are then further processed to remove duplicates arising from PCR artefacts using Picard v1.130 and to realign reads surrounding indels using GATK v3.3-0. Pileups are then created across all samples using SAMtools and SNPs are called using the consensus-method of BCFtools v1.1-137. The resulting SNPs were then filtered to remove those variants with phred-scaled quality score below 20, minor allele frequency below 0.01 or SNPs that were called in less than 10% of samples.

## RESULTS

### Simulation study

We first investigate the power and the false positive rate (FPR) of our method by simulations as minor allele frequency (MAF), sequencing depth, and sample size are varied. A total of 3,000 contexts per sample, of which one was a variant site with two possible alleles across the population, were simulated by sampling counts from a multinomial distribution. This corresponds to a simulating a SNP, indel or any other variant whose first base, i.e., the base directly following the context, is bi-allelic. Each context was simulated with a sequencing read error of 1% by sampling from a multinomial distribution, with the total number of simulations per context determined by the specified sequencing depth. Variants were determined by fitting a gamma distribution and rejecting at a level of $p < 0.05$ corrected for multiple testing by Bonferroni's method. This procedure is repeated 1,000 times for each combination of simulation parameters.

Figures 1 and 2 shows the results of the simulation. With a depth of 25 our method is able to recover the variant site with high power when the MAF is 20% or higher, even with few samples (50). The FPR was also well controlled, but reduces sharply with moderate depth (>25) at 100 samples, and is low at most depth for 1,000 samples. Identification of rare variants at low sample sizes (1% MAF at 100 samples) is not reliable, however rare variants are still identifiable with high power at high depth and samples (depth greater than 64 and 1,000 samples).

### Empirical experiments

We also evaluated our method on three different datasets: two datasets are of *Streptococcus pneumoniae* bacteria, one collected in Massachusetts (*Croucher et al., 2013*) and the other in Thailand (*Chewapreecha et al., 2014a*); and one mouse dataset (*Fairfield et al., 2011*). The two *S. pneumoniae* datasets comprise 681 and 3,369 samples sequenced using Illumina sequencing technology. The Jax6 mouse dataset (*Fairfield et al., 2011*) contains sequenced exomes of 16 inbred mouse lines.

All experiments were conducted with suffix-free contexts and only contexts present across all samples were evaluated for variants. Our method identified 40,071 variants in the Massachusetts dataset, 57,050 in the Thailand dataset, and 50,000 in the mouse dataset. We refer to these as *KL variants*.

We also compare our method with a mapping-based SNP calling approach on the *S. pneumoniae* datasets. Using sequence for *S. pneumoniae* ATCC 700669 (NCBI accession NC_011900.1) as a reference, there were 181,511 and 251,818 SNPs called for the

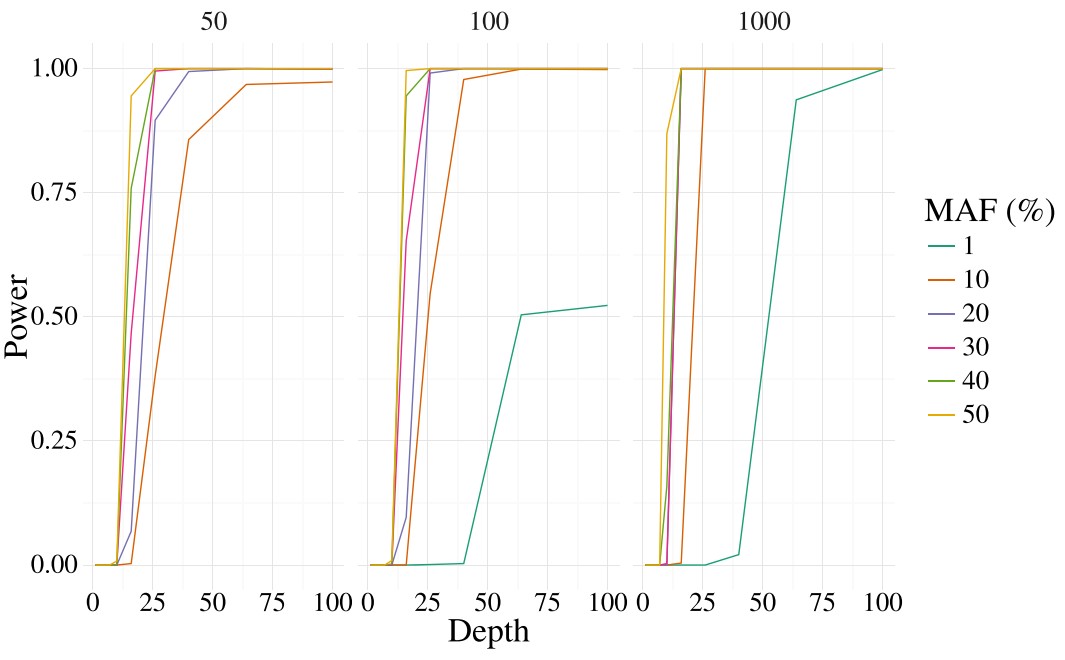

**Figure 1** **Power curves for 3,000 simulated contexts with a single variant context for varying depth and sample size (panels).** The bi-allelic variant context was simulated 1,000 times and curves show the mean of the 1,000 simulations. The error for the mean is less than 3% in all cases.

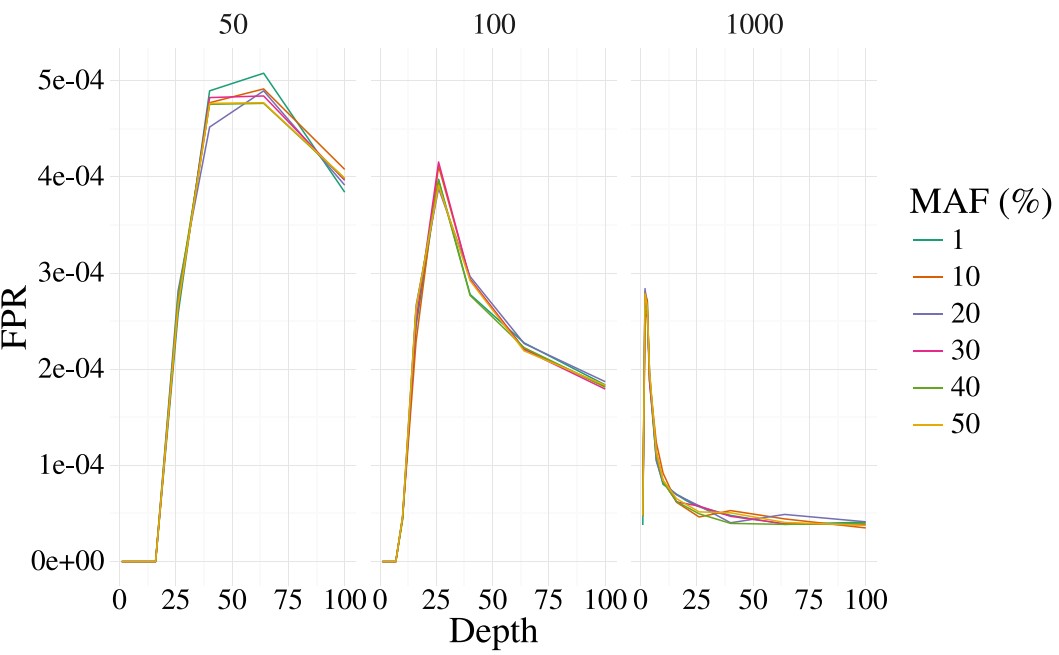

**Figure 2** **False positive rate for 3,000 simulated contexts with a single variant context for varying depth and sample size (panels) as described in Fig. 1.** The error for the mean is less than 3% in all cases.

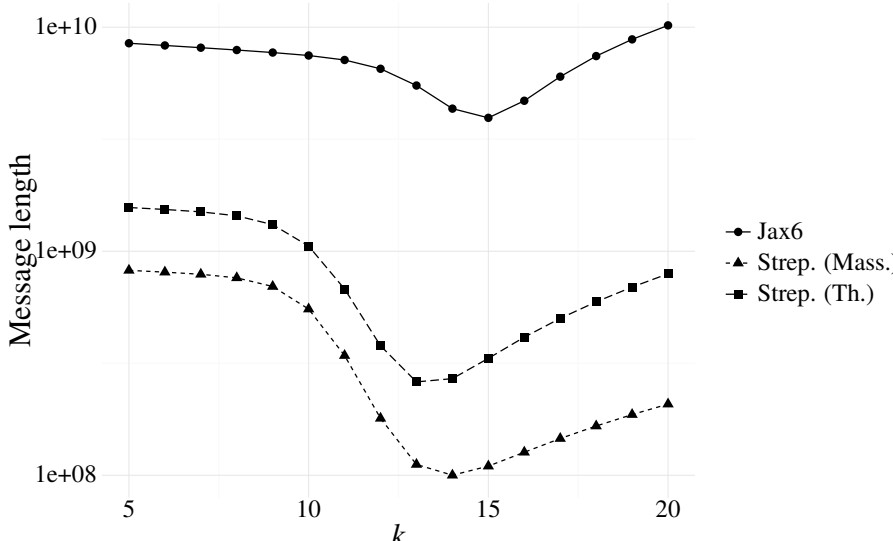

**Figure 3  Message length for prefix-only contexts on two *S. pneumoniae* samples from the Massachusetts and Thailand datasets, and the 129S1/SvImJ mouse line from the Jax6 dataset.** The optimal $k$ under the MML framework is $k = 14$ for the *S. pneumoniae* Massachusetts datasets, $k = 13$ for the *S. pneumoniae* Thailand dataset, and $k = 15$ for Jax6.

Massachusetts and Thailand datasets. To be comparable with the resulting binary SNPs calls, we transform our multi-allelic variants to binary variants with the major allele being one and other alleles being zero.

Finally, we compare our results with variants called by another reference-free caller DiscoSNP++ (*Uricaru et al.*, *2015*) (v2.2.1). DiscoSNP++ finds 8,728 variants for the Massachusetts *S. pneumoniae* data, and 290,615 variants for Jax6. DiscoSNP++ results are not available on the Thailand dataset as the software fails to run in reasonable time on such a large dataset.

## Message lengths

Our first experiment investigated the optimal $k$ resulting from our message length criterion (see 'Choosing context size'). Figure 3 shows the results of various contexts sizes on three samples, one from each of the Massachusetts *S. pneumoniae*, Thailand *S. pneumoniae* and Jax6 mouse data. The *S. pneumoniae* Massachusetts and Thailand samples had the shortest message length at $k = 14$ and $k = 13$ respectively, and the 129S1/SvImJ mouse line had the shortest message length at $k = 15$.

To evaluate the stability of the message length criterion, the optimal $k$ according to message length was calculated on all samples from the Massachusetts data (Table 1). The majority of samples had an optimal length of $k = 14$, with the remainder being optimal at $k = 13$. Investigation into the singleton sample with minimal length at $k = 8$ revealed a failed sequencing with only 18,122 reads present. We also evaluated all samples present in the Jax6 dataset and found all but one samples had minimal message length at $k = 15$. The stability of $k$ is therefore high and we use $k = 14$ for the two *S. pneumoniae* datasets and $k = 15$ for the Jax6 mouse dataset henceforth in all experiments.

**Table 1** Count of samples in Massachusetts data by optimal *k*.

| Optimal *k* | Count |
|---|---|
| 8 | 1 |
| 13 | 304 |
| 14 | 376 |

**Table 2** AMR prediction results using KL variants. Variants were discovered only on the Massachusetts dataset and then called on both Massachusetts and Thailand datasets. Each row indicates what dataset models were trained on and the columns denote the testing dataset. Numbers are the Area Under the Receiver Operating Characteristic (AROC). The AROC was estimated using 10-fold cross-validation within datasets. The numbers in parentheses are the performance when predicting on standard SNP calls (S) derived through a traditional alignment pipeline and DiscoSNP++ (D) calls. DiscoSNP++ results are not available on the Thailand dataset as the software fails to run in reasonable time on such a large dataset.

| Training dataset | Massachusetts | Thailand | All |
|---|---|---|---|
| Massachusetts | 95.6 (S: 94.4, D: 96.6) | 81.3 (S: 88.6) | |
| Thailand | 72.5 (S: 66.8) | 97.6 (S: 97.6) | |
| All | | | 97.1 |

## Supervised learning performance

To investigate the robustness of our variants for genomic prediction tasks, we evaluated the ability of variants called on the Massachusetts *S. pneumoniae* dataset for the prediction of Benzylpenicillin resistance under different training and testing scenarios across the two *S. pneumoniae* datasets. Each sample was labelled as resistant if the minimum inhibitory concentration exceeded 0.063 µg/mL (*Chewapreecha et al., 2014b*). In all tasks, a support vector machine (SVM) (*Schölkopf & Smola, 2001*) was used to predict resistance from the variants, and the performance measured using the Area under the Receiver Operating Characteristic (AROC).

Table 2 shows the results of the experiments. Each row indicates what dataset models were trained on and the columns denote the testing dataset. For intra-dataset experiments (i.e., the diagonal), AROC was estimated using 10-fold cross validation.

Our variants are clearly capturing the various resistance mechanisms, as evident by the strong 10-fold cross validation predictive performance. In comparison to the traditional pipeline and DiscoSNP++ features (on Massachusetts data only) also performed well. Given the high level of accuracy, the three methods do not differ significantly in performance.

The model trained using our variants on the Massachusetts data is moderately predictive on the Thailand dataset. Conversely, the model from the Thailand dataset can also moderately predict resistance in the Massachusetts data, but to a lesser degree. One possible explanation for this limited predictive ability is the existence of resistance mechanisms unique to each dataset, hence a model trained on one dataset will not capture unobserved mechanisms and consequently the model is unable to predict resistance arising form these unknown mechanisms. This hypothesis is supported by the strong performance observable on the diagonal: when combining both datasets and preforming cross-validation, the performance is high.

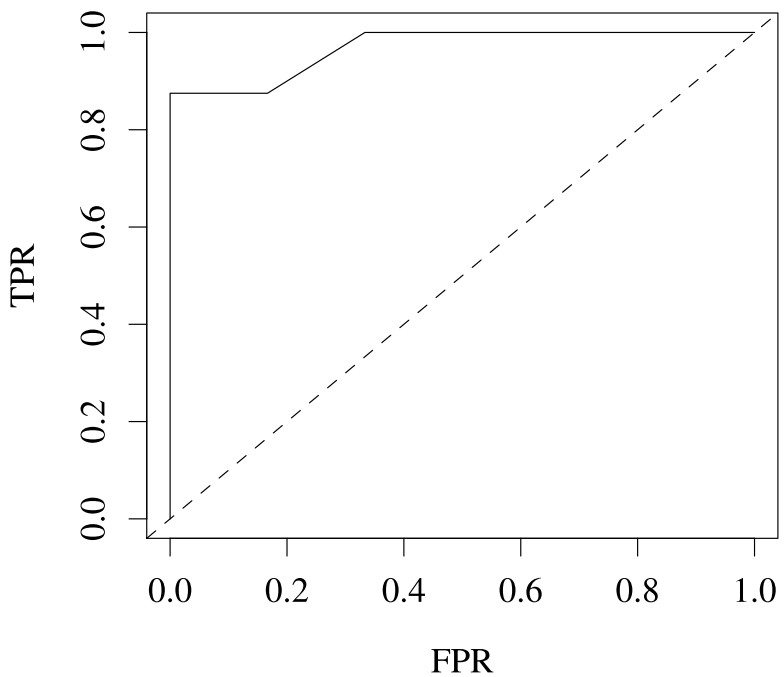

**Figure 4** **ROC produced from leave-one-out cross-validation performance predicting agouti coat colour from KL variants on Jax6 mouse dataset.** AROC is 96%.

We also evaluated our variants for predicting coat colour on the Jax6 mouse dataset (*Fairfield et al., 2011*). As few samples are available (14 labelled samples), we reduced the problem to a 2-class classification problem, classifying coat colour into agouti or not. This led to a well balanced classification problem with 8 samples in the agouti classes and 6 not. The performance for this task was estimated at 96% AROC using leave-one-out (LOOCV) cross-validation, suggesting the variants are also predictive of heritable traits in higher level organisms. Fig. 4 shows the ROC for this classification problem.

## Population structure

Finally, we investigate the population structure captured by KL variants and the SNP calls on the Massachusetts dataset. The population structures were estimated using Principle Component Analysis (PCA), a common approach whereby the top principal components derived across all genetic variants reflect underlying population structure rather than the studied phenotype of interest (*Price et al., 2010*). Five sub-populations (clusters) were identified using $k$-means on the first two principal components from the SNP data. Projecting those 5 clusters on to the principal component scores of our variants (Fig. 5) results in highly concordant plots. Four out of the five clusters can be easily identified using our variants, indicating the detected variation preserves population structures well.

A canonical correlation analysis (CCA) was performed to further assess the similarities between the two feature sets (Table 3). Regularisation was used to find the canonical vectors as the cross-covariance matrices are singular for our dataset. As there are significantly more features than samples, regularised CCA was used and the correlation between projections estimated using 100 samples of leave-one-out bootstrap (*Hastie, Tibshirani & Friedman*,

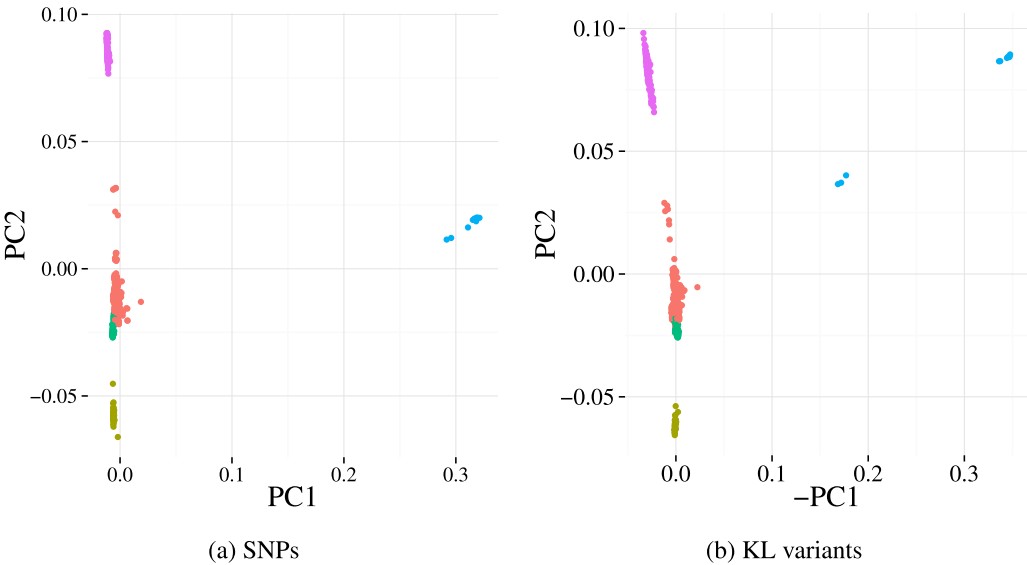

(a) SNPs                              (b) KL variants

**Figure 5** **First two principal components derived from alignment-based SNP calls (A) and from variants detected by our method (B) applied to the Massachusetts *S. pneumoniae* dataset.** Each point represents a sample and the colours denotes the cluster assignment determined by $k$-means clustering. The similar pattern of samples in each plot indicates that the same population structure signal is detected by the two variant detection methods.

**Table 3** **Correlation coefficients for first 5 CCA components, estimated using 10-fold cross-validation on Massachusetts data.**

| Component | Correlation coefficient ($\pm$95% CI) |
| --- | --- |
| 1 | $0.873 \pm 0.014$ |
| 2 | $0.880 \pm 0.006$ |
| 3 | $0.877 \pm 0.007$ |
| 4 | $0.862 \pm 0.007$ |
| 5 | $0.867 \pm 0.008$ |

*2013*). We found the first three components explain all the variance (99%), with the first component alone explaining 76%. Therefore, both mapping-based SNPs and KL variants are largely capturing the same variance on the Massachusetts data.

## Analysis of contexts

To further elucidate the type of variants that are being discovered by our method, we aligned the significant contexts from the Massachusetts dataset to the *S. pneumoniae* reference. Of the contexts, less than 1% failed to align, 41% aligned in a single location, and the remainder aligned in two or more locations.

One context aligned in 82 different locations in the reference genome. Further investigation revealed the context corresponds to a *boxB* repeat sequence. Such repeats have previously been used to identify population structure of *S. pneumoniae* isolates carrying the 12F serotype, supporting our population structure findings (*Rakov, Ubukata & Robinson, 2011*). This suggests the variants may be tagging more complex structural elements than just single nucleotide variants.

## CONCLUSIONS

We presented a novel reference-free variant detection method for next-generation sequence data. Our method has the advantage of no tuning parameters, rapid calling of known variants on new samples, and may be suited for targeted genotyping once a known set of variants are obtained.

Simulation experiments showed the method is relatively robust and has good power and FPR to detect common variants, but for rare variants the power was lower and a high depth and number of samples were required to reliably detect them.

In a typical genomic prediction setting the method was able to predict heritable phenotypes on both a bacterial dataset (anti-microbial resistance) and on a mouse dataset (coat-colour). On the *S. pneumoniae* datasets, our method was shown to have similar performance to a standard alignment-based SNP calling pipeline, with its requirements for a suitable reference genome. Moreover, the method was shown to capture the same population structure on the Massachusetts Streptococcus bacterial datasets as an alignment-based variant calling approach. These results show our method is capable of capturing important genomic features without a known reference.

As with other reference-free variant calling methods, interpretation of the detected variants is more difficult compared to a mapping-based approach as called variants are reported without positional information. One approach to obtain such annotations is to map the variant and its context back to a given reference. Given that most sequences with a length greater than 15bp that exist in a given bacterial reference will have a unique mapping, many variants could be easily mapped back. However, such information is unlikely to exist for variants that do not occur in the reference, or may be misleading for variants that arise through complicated procedures such as horizontal gene transfer. Alternatively, variants and their context could be examined via BLAST searches to determine whether these sequences correspond to previously identified genes or other genomic features.

In our experiments we used a combination of these approaches to investigate some of the variants found on the bacterial dataset. We identified contexts that mapped to numerous locations in the reference genome and then used BLAST to identify the likely origin of the sequence. Through this method, variants associated with boxB repeat sequence were found, suggesting our method is capturing variance associated with complex structures.

We envisage that the method proposed here could be used to conduct a rapid initial analysis of a given dataset, such as species identification, outbreak detection or genomic risk prediction. Our method also enables analysis of data without a suitable reference while still avoiding the computationally expensive step of assembly. Furthermore, our method scales linearly with the total number of reads, allowing application to large datasets.

The statistical framework established in this work is quite general and could be expanded in several ways. While we have examined only single nucleotide variants within this work, insertions and deletions could be explicitly modelled within this framework at the cost of increased computational expense. It may also be possible to model other types of variants, such as microsatellites, provided that a suitable representation for them could be found.

## ACKNOWLEDGEMENTS

We thank Thomas Conway and Noel Faux for helpful discussions.

### Funding

The authors received no funding for this work.

### Competing Interests

The authors declare there are no competing interests.

### Author Contributions

- Justin Bedo conceived and designed the experiments, performed the experiments, analyzed the data, wrote the paper, prepared figures and/or tables, performed the computation work, reviewed drafts of the paper.
- Benjamin Goudey performed the experiments, analyzed the data, wrote the paper, reviewed drafts of the paper.
- Jeremy Wazny performed the experiments, wrote the paper, reviewed drafts of the paper.
- Zeyu Zhou performed the experiments, analyzed the data, wrote the paper, prepared figures and/or tables, reviewed drafts of the paper.

### Data Availability

The research in this article did not generate any raw data and all experiments were conducted on publicly available data.

The Jax6 mouse exome data is available from http://phenome.jax.org/db/q?rtn=projects/projdet&reqprojid=466.

The bacterial isolates are available from the SRA. The accessions are available in Supplemental Information 1.

### Supplemental Information

Supplemental information for this article can be found online at http://dx.doi.org/10.7717/peerj-cs.71#supplemental-information.

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
