# Peer review of "Information theoretic alignment free variant calling"

_PeerJ Computer Science, doi:10.7717/peerj-cs.71_

## Round 0.1 · original submission · Major Revisions

Dear authors

The second referee is asking for major revisions to enforce experimental validation.

It is especially important that the question about a possible bias in the method is answered properly. The answer might arise from an additional run on new data.

A second important requirement is the need for control data.

Reviewer 1 ·

Basic reporting

No Comments

Experimental design

The article under review is more related to biology than computer science. Thus, I am not sure whether PeerJ Computer Science is the most suitable place for the article.

Validity of the findings

No Comments

Additional comments

The article proposes an alignment and assembly free variant calling method which is based on the context surrounding particular nucleotides. It consists of two steps: variant calling and variant finding. In order to find the length of the context, information theoretic principles are used.

The article is well-written. The new method is compared with previous ones and it seems quite effective. My only concern is that the paper seems to be more relevant for biology and thus, I am not sure whether PeerJ Computer Science is the most suitable place for publication.

I conclude with three (minor) typos the authors might want to correct in a revision:

Line 201: "...is therefore high and we used...";

Line 234: appli -> apply?

Line 272: delete one of the two "avoiding".

·

Basic reporting

Just to fix the main ideas, this paper describes and justifies a method aiming to identify genetic variants inside a species, given a set of genomic sequences. These variants are not restricted to be SNPs only, but can result from INS/DEL modifications as well. Furthermore, and certainly more importantly, this method has been designed to be self contained, i.e. without reference to any a priori known test genome. This problem is therefore of great interest and any certified solution will be of great use.

Basic reporting
The paper is generally well written and easy to read ; the mathematical framework is not too exotic and the references allow the reader to go easily to omitted proofs or developments. One will however be surprised that the authors use systematically “statistic” and not “statistics” as a substantive, whereas it is declared as an adjective in most dictionaries.
As mentioned previously literature is well referenced and relevant.
However the abstract and introduction do not show properly the goals of this study, its major achievements and the domain of validity of the method. This is a real weakness of the paper.
Figure 1 is good, Figure 2 average but Figure 3 is weak and hardly understandable.

Experimental design

Experimental proof is weak. Only three sets of genomes have been study which is clearly not enough for validation.

Validity of the findings

As I said in my intro, the problem is a real one and a good solution will have a real impact. The mathematical background in statistics is neither original nor very hard, and there is no really new nor deep theorem, which does not bother me: the originality can come from having the idea of building evidence from already known features.
My main problem comes from the fact that I do not see a link between the mathematical properties and the claim that this method will indeed find the relevant variants in most situations. The last paragraph of the introduction (lines 52-57) is apparently based on 3 experimental studies with no quantitative argument! It is surprising that the numbers of variants are all of the same orders of magnitude (lines 176-179) and that there is no analysis of a possible biais in the method under certain conditions.
The last paragraph of the introduction in Section 3 (lines 186-189) is also puzzling: any computer scientist should explain why a program is so time consuming…
Finally, I think that the considerations about the accuracy of the predictions should be stated by more rigorous experimental checking. I would guess that the method goes not so badly whenever the solution is close to the center of the gaussian, but that it works poorly elsewhere.
A first check would be to test the accuracy would be to start with the prefix/suffix model on validated data under several different genetic situations. Then I would try INS/DEL situations again under different clustering conditions.

I think that it should then be possible to argue more convincingly on the validity of the method, since a formal mathematical proof is very likely out of scope.

Additional comments

I suggest that the authors strengthen their experimental validation by a more thorough and systematic evaluation of the relevance of their results. Paragraphs such as 2.4, 2.5 and Section 3 should be rewritten. The introduction and the conclusion will be accordingly modified.

---

## Round 0.2 · accepted · Accept

Thank you for the additional simulation study on page 6, and your analysis of contexts page 12. I have now reviewed your rebuttal and revision and the article is Accepted.